# Challenges and Opportunities Presented by Current Techniques for Detecting Schistosome Infections in Intermediate Host Snails: A Scoping Review

**DOI:** 10.3390/ijerph18105403

**Published:** 2021-05-19

**Authors:** Onyekachi Esther Nwoko, John J. O. Mogaka, Moses John Chimbari

**Affiliations:** Discipline of Public Health Medicine, College of Health Sciences, University of KwaZulu-Natal, Durban 4000, South Africa; johnmogaka2@gmail.com (J.J.O.M.); chimbari@ukzn.ac.za (M.J.C.)

**Keywords:** intermediate host snails, schistosomiasis, diagnostics, conventional, immunological, nucleic-acid amplification, eDNA

## Abstract

Schistosomiasis, a neglected tropical disease (NTD), causes morbidity and mortality in over 250 million people globally. And 700 million people are at risk of contracting it. It is caused by a parasite of the genus *Schistosoma*. Freshwater snails of the family Planorbidae are of public health significance as they are intermediate hosts of these highly infective flukes. Accurate diagnostic techniques to detect schistosome infections in intermediate host snails (IHS) and environmental surveillance are needed to institute measures for the interruption of transmission and eventual elimination. We carried out a systematic review of the literature to assess advantages and limitations of different diagnostic techniques for detecting schistosome infections in snails. Literature from Scopus, Web of Science, and PubMed databases from 2008 to 2020 were searched using combinations of predefined search terms with Boolean operators. The studies revealed that conventional diagnostics are widely used, although they are labor-intensive, have low specificity and sensitivity levels, and cannot detect prepatent infections. Whereas more advanced techniques such as immunological, nucleic-acid amplification, and eDNA diagnostics have high sensitivity and specificity levels, they are costly, hence, not suitable for field applications and large-scale surveys. Our review highlights the importance of designing and developing innovative diagnostics that are high in specificity and sensitivity as well as affordable and technically feasible for use in field laboratories and for large-scale surveys.

## 1. Introduction

Parasitic organisms, the causative agents of some of the most neglected but prevalent infections, constitute endemic and emerging public health threats in many parts of the world, mainly in poor and developing countries with poor sanitation and limited health care [1]. This group of pathogens includes the protozoans *Trypanosoma* (African sleeping sickness), *Leishmania* (leishmaniasis), *Plasmodium* (malaria), and the helminths such as *Schistosoma* (schistosomiasis). They are geographically widespread and affect vulnerable and non-vulnerable human populations due to their high prevalence and the number of deaths they cause each year. Of these parasites, schistosomiasis is the most important water-based disease from a global public health perspective [2]. It is caused by trematode parasites of the genus *Schistosoma*, of which three major species are of medical significance in human health: *Schistosoma haematobium* (occurring in Africa and the Arabian Peninsula), *Schistosoma mansoni* (endemic to Africa, the Arabian Peninsula, South America, and the Caribbean) and *Schistosoma japonicum* (restricted to China, the Philippines, and Indonesia). *Schistosoma mekongi*, in the Mekong River basin of Southern Laos and northern Cambodia, and *Schistosoma guineensis* and *Schistosoma intercalatum* in West and Central Africa, are locally distributed species that also cause human disease. Interspecies infections with hybrids of *S. haematobium*, *S. bovis*, *S. curassoni*, and *S. mattheei* have occurred in Corsica, France, and some West African countries [3].

Several climatic, geographic, social, and economic issues present risks for contracting schistosomiasis they modulate the prevalence and incidence of the schistosomes [4]. For instance, human economic development, especially of water resources in the form of large dams and irrigation systems, present one of the key risk factors for the spread and intensification of schistosomiasis [5,6]. Climate change has been identified as a risk factor in escalating the transmission or extension of schistosomiasis to areas where the disease previously was not endemic. [7,8,9,10,11,12].

Although sustained and ambitious goals aimed at controlling schistosomiasis have been set by governments in collaboration with multilateral agencies such as the World Health Organization (WHO), the disease persists [13,14]. This is partly due to over-reliance on large-scale drug treatment (preventive chemotherapy—PCT) as well as a lack of accurate diagnostic tools for case detection and community screening in areas where the disease is endemic [15]. Despite the mass and targeted administration of drugs such as the highly efficient praziquantel (PZQ), there is evidence that schistosomiasis is gradually becoming endemic in new areas [16]. Although schistosomiasis is effectively treated with PZQ, rapid reinfection with rebound morbidity precludes effective control based on PCT alone. This has necessitated that WHO encourage elimination of infections due to schistosomiasis and reduction of heavy infection intensities in at-risk populations [17] through the combination of PCT and other interventions, such as improved access to clean water, sanitation, and hygiene (WASH), information, education, and communication (IEC), and snail control [18].

Freshwater snails of the family Planorbidae are of public health significance as they are intermediate hosts of the infective Schistosoma parasites. Snail control is aimed at interrupting the schistosomiasis transmission chain through a variety of methods, such as environmental, biological, and chemical controls. However, to undertake snail control as a means of eliminating infections due to schistosomiasis, accurate, reliable, and feasible methods for the diagnosis of the etiological agents must be employed. To understand the transmission, control, and eventual elimination of schistosomiasis, accurate identification of schistosome species infecting intermediate host snails is imperative. Effective schistosome diagnosis plays a key role in the control and elimination strategies, with wide applications in both high- and low-prevalence case detections. Diagnostics with high specificity and sensitivity are required to promote transmission interruption, leading to control and elimination. Currently, the diagnosis and identification of schistosomes that infect humans are done using several techniques, such as morphology, microscopy, and molecular assays. Each method has both advantages and disadvantages. Furthermore, many research laboratories are developing new diagnostic methods as well as improving old ones for natural or experimental parasite detection in snail intermediate hosts. Many of the current diagnostic techniques have practical limitations despite technical improvements. For instance, cercariae emerging from infected snails cannot be precisely identified morphologically to the species level or as hybrids, hence, molecular means of pathogen diagnosis can be used to identify the schistosome species shed by intermediate host snails (IHS). Even though there has been a focus on the development of novel molecular diagnostic techniques, their application in field settings is limited unless the tests are inexpensive and field-deployable, as schistosomiasis is generally endemic to resource-poor settings.

The objective of this review was to identify and describe the advantages and disadvantages of different diagnostic techniques for detecting natural and/or experimental schistosome infections in IHS. Specifically, (i) we reviewed the prominent and promising techniques of schistosome cercariae detection, (ii) discussed analytical pitfalls and bottlenecks of the techniques, (iii) provided contextual considerations in technique selection, and (iv) highlighted knowledge gaps regarding the relevance of the techniques in the control of schistosomiasis. We systematically searched for articles published about schistosome detection and identification in IHS in Africa in both national and international journals from 2008 to August 2020. We acknowledge that this review may not cover all schistosome identification methods as we could only rely on what was in the public domain and written in English.

## 2. Materials and Methods

### Search Strategy and Selection Criteria

A systematic search for literature from Scopus, Web of Science, and PubMed databases was conducted. Studies that reviewed techniques used in detecting infection in intermediate host snails for human schistosomiasis in Africa were considered using the following terms and Boolean operators (OR, AND): Snail crushing OR snail squashing OR snail smashing, Snail dissection OR snail shell removal, Cercariae shedding OR cercarial shedding, Histological examination OR Biochemical techniques OR enzymatic electrophoresis, Molecular biology techniques OR molecular approaches OR molecular methods OR RNA blotting OR DNA blotting OR classical polymerase chain reaction (PCR) OR multiplex PCR OR real-time PCR OR quantitative PCR OR nested PCR OR Reverse Transcription PCR OR assembly PCR OR Loop mediated isothermal amplification (LAMP) OR Reverse Transcription Loop-mediated Isothermal amplification (RT-LAMP), Infection OR infection detection OR intensity, schistosomiasis OR Bilharzia OR *Schistosoma mansoni* OR *Schistosoma haematobium* OR Schistosoma, intermediate host snails OR snail intermediate host OR intermediate host OR freshwater snails OR snail host OR snail vector OR malacology OR Biomphalaria OR Bulinus OR Bulinid. The review focused on addressing the following questions: What techniques are used to detect schistosomiasis infection in intermediate host snails and what are the advantages and limitations of each technique with regards to sensitivity, specificity, cost, and ease of use?

A country filter was applied to limit the studies to Africa. Articles that highlighted techniques for detecting infections in IHS were included while articles that focused on the effects of selected plants on schistosomiasis in intermediate host snails were excluded. Studies that involved mice being infected in a water body and taken to the laboratory for monitoring were also included. Only articles published from 2008 to 2020 were included in the study. Papers earlier than 2008 were excluded as they were dealt with in an earlier review by Caron et al. [19].

A total of 278 articles were obtained from 3 databases. Eighty-three duplicate articles that were identical in Scopus, Web of Science, and PubMed were removed. Ninety-eight articles were also excluded after title and abstract screening because they did not highlight techniques for detecting human schistosomiasis infection in intermediate host snails. Seventy-three articles were further excluded because they focused on the molecular diversity, variability, and susceptibility of snail species rather than detection methods, and detailed descriptions of the same. Reference lists of selected papers were checked to identify additional studies for inclusion in the study. A total of 23 articles were fully reviewed. Our review was guided by the Preferred Reporting Items for Systematic Reviews and Meta-Analyses (PRISMA) [20] (Figure 1). Our discussion applied WHO’s Affordable, Sensitive, Specific, User-friendly, Rapid and robust, Equipment-free and Deliverable (ASSURED) to end-users criteria [21]. 

## 3. Results

### 3.1. Overview of Selected Studies

The articles we fully reviewed examined techniques used in detecting infections in schistosomiasis intermediate host snails in 12 African countries. The studies were done in Tanzania (5), Egypt (4), Nigeria (4), Kenya (3), Chad (1), Ethiopia (1), Morocco (1), Senegal (1), South Africa (1), Sudan (1), Uganda (1), and Zimbabwe (1). Four themes of schistosome detection techniques were identified from the 23 articles: (1) conventional diagnostics, (2) immunological diagnostics, (3) nucleic-acid amplification diagnostics, and (4) environmental DNA (eDNA) based diagnostics. Of the 23 articles reviewed, 3 articles concerned IHS that were experimentally infected, 1 article concerned IHS that were both experimentally and naturally infected, and 20 articles concerned naturally infected IHS. Sixteen articles [22,23,24,25,26,27,28,29,30,31,32,33,34,35,36,37] used conventional diagnostics to detect schistosome infection. The second most-used diagnostic technique was nucleic-acid amplification, used in 13 articles [22,23,24,25,26,27,33,34,38,39,40,41,42,43]. Three articles used eDNA-based diagnostics [36,37,44] while 1 article used the immunological diagnostics [41]. None of the articles applied all the techniques for detecting infection in intermediate host snails simultaneously. However, 10 articles used 2 techniques simultaneously to detect schistosomes. Seven out of 10 of those articles addressed conventional and nucleic-acid amplification diagnostics [22,23,24,26,27,33,34] while 2 used conventional and eDNA diagnostics [36,37]. One of the 10 articles used immunological and PCR-based molecular diagnostics [41]. Table 1 presents information extracted from the 23 surveyed articles that included author/reference, snail species, technique(s) and themes identified, objectives, outcome, advantages, limitations and country.

### 3.2. Theme 1: Conventional Diagnostics

This is one of the categories of diagnostics used in detecting schistosome infections in IHS. The techniques are based on using the naked eye with the aid of a hand lens or microscopy. The three techniques under this diagnostic category are cercariae shedding, snail crushing, and dissection.

#### 3.2.1. Cercariae Shedding

Snail shedding is the traditional method used to detect schistosome cercariae that emerge from an infected snail host [24,25,28,29,31,35,38]. Once miracidia infect snails, they undergo development within the snail host and upon maturing, are shed into the surrounding water as cercariae. Typically, cercariae emerge in greatest numbers during the day under sunshine, although minimal numbers of cercariae will still emerge in the dark. Exposing an infected snail to light of enough intensity triggers them to release cercariae. Therefore, in controlled laboratory settings, investigators can artificially manipulate light intensities to take advantage of maximal release of cercariae at a time of the investigator’s choice. This classical method usually begins with obtaining snail samples by scooping or handpicking, using a counts-per-unit-of-time sampling method. Collected snails are rinsed and placed individually in test tubes or Petri dishes containing filtered water with a near-neutral pH at room temperature (25 to 30 °C). Each snail is exposed to indirect sunlight or electric light with alternating periods of light and darkness for 1 h to induce cercariae shedding. If a snail sheds cercariae, they will be visible as tiny specks suspended or wriggling in the water (swimming) when the tube is held against the light and a 10 × hand lens is useful to view them. Alternatively, the wells of the tubes or Petri dishes can be examined for the presence of cercariae under a dissecting or stereoscopic microscope. Snails that would have not shed cercariae on the first exposure are kept in glass aquaria in the laboratory and rechecked for cercariae shedding for 4 weeks. Absence of cercariae shedding at the 4th week of exposure to sunlight is indicative of the absence of infection. Cercariae types can be identified to the genus level based on visual morphological characterization using standardized taxonomic keys. The frequency of the different types of cercariae observed is then recorded, and overall parasitic infection rates can be calculated.

Bukuza, et al. [24] used the cercarial shedding technique to detect infected schistosomiasis snails in Gombe, western Tanzania and obtained an infection rate of 12%. Luka et al. [28] found a 1.75% and 0.55% infection rate among *Bulinus globosus* and *Lymnae natalensis*, respectively, using the cercarial shedding approach for detecting schistosomiasis infection in Borno state, Nigeria. Mereta et al. [29] found 3.6% snails infected with cercariae. *B.pfeifferi* was found to be the most highly infected snail, accounting for about 85% of all infected snails in Omo Gibe river basin, Southwest Ethiopia. Ejehu et al. [25] investigated the infection of schistosomiasis intermediate host snails in Lake Oguta 1 region, Imo state, Nigeria using the cercarial shedding technique. However, out of 385 snails collected, none shed cercariae. Senghor et al. [35] evaluated the infection of schistosomiasis intermediate host snails and the impact of drought on the snails in the Niakhar region of Fatick, west-central Senegal using the cercarial shedding approach. The overall infection rates were 0% in 2012 and 0.12% in 2013 for *Bulinus senegalensis* and 13.79% and 4.98% in 2012 and 2013, respectively, for *Bulinus umbilicatus*. Moser et al. [31] used cercarial shedding to study the infection rates of schistosomiasis intermediate host snails in N’Djaena, Chad. Infection rates of 0.84% and 0.93 were recorded among *Bulinus truncatus* and *Biomphalaria pfeifferi*, respectively, in the dry season. Opisa et al. [32] analyzed the infection rate of schistosomiasis intermediate host snails in Kisumu city, western Kenya with an infection rate of 1.8% using cercarial shedding. Mohammed et al. [30] obtained an infection rate of 14.1% in schistosomiasis intermediate host snails in the East Nile locality, Khartoum, Sudan using the cercarial shedding approach. Allan et al. [23] obtained an infection rate of 3.96% using cercarial shedding to detect infected schistosomiasis intermediate host snails. Farghaly et al. [26] compared the sensitivity of cercarial shedding used to detect *Schistosoma mansoni* infection in *Biomphalaria alexandrina* snail host to PCR assay. The average sensitivity of the cercarial shedding method to PCR was 23.8% while the sensitivity of PCR was 100%.

The study of infected snails capable of shedding cercariae is the most common and practical method to assess human-snail disease transmission rates, as well as estimate the risk of infection in transmission sites. However, the number of shed cercariae is not as large as can be observed when infected snails are crushed, as well as when affected snail tissues are dissected.

#### 3.2.2. Snail Crushing

Aboelhadid et al. [22], Saad et al. [33] and Farghaly et al. [26] used the snail crushing method to detect *Schistosoma* spp. infection in snails. In snail crushing, the snails are placed in an inverted plastic Petri dish top, then their shells are gently crushed by placing the bottom of the dish on top of the snails and pressing down. Water is added to the crushed snail and secondary sporocysts teased out from the tissue are placed in small (50 mm) Petri dishes with water. Upon teasing the sporocysts apart, cercariae are sporadically released; the most infective (mature) cercariae are those that swim to the top of the water and hang there. The snail’s shell and viscera are examined under a dissecting microscope to detect sporocysts and cerceriae. Farghaly et al. [26] compared the average sensitivity of the snail crushing method to PCR. It was found to be 46.4%.

#### 3.2.3. Snail Dissection

In snail dissection and cercariae morphometrics, the flesh of the snail is carefully removed from the shell and centrifuged. The supernatant is collected and examined under a microscope [21]. Motility and morphology of live, unstained cercariae that emerge from the snails are noted and classified to the major type levels [45]. The cercariae body shapes can be used for cercariae identity. Under a microscope, body shapes may be classified as being either oval, cylindrical, or flat dish shaped. The cercariae are also identified and grouped according to body coverings, length of body, tail shafts, and forked tail [46]. The movement behavior of cercariae may also be used to identify and group them: i.e., whether the cercariae floats on the surface or in the water.

Fixation of the soft snail tissues for microscopic examination is a useful technique for identifying schistosome cercariae under a dissecting microscope. The successive intra-molluscan stages of schistosome development are called primary (mother) and secondary (daughter) sporocysts. Both primary and secondary sporocysts may be detected after they develop in the snails, and just before cercariae begin to emerge from infected snails. Under proper fixation conditions, the primary sporocysts can be seen easily as white masses, or swellings, usually in the head-foot and/or tentacles of the snail that was exposed to miracidia 10–14 days previously.

### 3.3. Theme 2: Immunological-Response Based Cercarial Detection Diagnostics

El Einin et al. [41] assessed the sensitivity and specificity of two methods; sandwich ELISA and nested polymerase chain reaction (nPCR) in infected *Biomphalaria alexandrina* snails by detecting *S. mansoni*. They compared the efficacy of each method to the conventional diagnostics. The sandwich ELISA approach has higher specificity than the conventional approach and was capable of detecting prepatent infections in *B. alexandrina* snails. However, the nPCR technique revealed higher sensitivity than the sandwich ELISA approach and detected *S. mansoni* infection by 100%.

### 3.4. Theme 3: Nucleic-Acid Amplification Diagnostics

The techniques under this diagnostic category are classical PCR, nested PCR, multiplex PCR, real-time quantitative PCR, molecular barcoding, and loop-mediated amplification (LAMP).

#### 3.4.1. Classical PCR

The classical PCR-based molecular technique is sensitive and specific in detecting schistosome parasites in intermediate host snails. Some studies in this review used this technique to detect and characterize the species of cercariae sheds.

Bakuza et al. [24] used the classical PCR and conventional diagnostic methods to determine the distribution and infection rate of *S. mansoni* in the Gombe area along the shores of Lake Tanganika in western Tanzania. An infection rate of 12% and 47% was observed from the conventional diagnostic and classical PCR, respectively. In addition, Allan et al. [23] compared the infection rate of *B. globosus* in Zanibar using the conventional diagnostic and classical PCR. An infection rate of 3.96% was observed from the conventional diagnostic whereas a consistently higher infection rate of 40–100% was detected with the classical PCR diagnostic. Moreover, Aboelhadid et al. [22] used the classical PCR and conventional diagnostics to compare their ability in detecting infection of snails by *S. mansoni*. Classical PCR assay detected schistosomes in infected snails while the conventional diagnostic failed to detect infections. Furthermore, Farghaly et al. [26] compared the sensitivity of the conventional diagnostic (cercarial shedding and snail crushing) used to detect *S. mansoni* infection in *B*. *alexandrina* snails to the classical PCR. Lastly, the average sensitivity of cercarial shedding to PCR was 23.8% and the average sensitivity of snail crushing to PCR was 46.4% while the sensitivity of PCR was 100%. Akinwale et al. observed an infection rate of 29.7% using the classical PCR in south-west Nigeria [39].

#### 3.4.2. Nested PCR

The nested polymerase chain reaction (nPCR) is a modification of the standard PCR protocol intended to address non-specific binding in PCR products due to the amplification of unexpected (unwanted) primer binding sites. Nested PCR is a two-step PCR cycle that employs two sets of primers. The first set of primers are designed to anneal to sequences upstream from the second set of primers and are used in the initial PCR reaction. The first PCR reaction produces amplicons that are then used as a template for a second set of primers and a second amplification step. Okeke et al. [42] employed the PCR method to detect schistosomes and the nPCR method to characterize the schistosomes. Seventy-seven percent of the snails found were positive for schistosome infection, but only 9.67% of the snails were positive for *S*. *mansoni* while the authors did not specify the species of the other infections.

#### 3.4.3. Multiplex PCR

In classical PCR, a single target is amplified in a single reaction tube. In comparison, multiplex PCR allows for simultaneous amplification of multiple target sequences in a single tube using specific primer sets in combination with probes labeled with spectrally distinct fluorophores. Multiplex PCR requires that primers lead to amplification of unique regions of DNA, both in individual pairs and in combinations of many primers, under a single set of reaction conditions. This means that one reaction includes many separate PCR reactions that result in the differentiation of each PCR amplicon and simultaneous measurement of expression levels of multiple target sequences of interest. The ability to simultaneously analyze multiple target sequences in a single tube allows for maximized use of precious starting material, such as human or animal tissue samples, as well as reduced reagent use. With appropriate optimization, this can translate to improved time- and cost-efficiency, yielding more data from each reaction and utilizing fewer reagents. Multiplex PCR is becoming a rapid and convenient screening assay in both the clinical and research laboratory. Schols et al. [34] and Saad et al. [33] used this method to detect and characterize infections.

#### 3.4.4. Real-Time Quantitative PCR

Quantitative PCR (qPCR) is a later modification of the original PCR protocol meant to address one of the drawbacks of the PCR process. Classical PCR first amplifies a DNA sequence and then analysis of the amplicon is done in retrospect. This means that quantification of the PCR product is extremely difficult because the PCR results in the same amount of product regardless of the initial amount of DNA template. This limitation was resolved in 1992 by connecting the PCR thermal cycler to a spectrofluorometer [47] through a technique known as real-time quantitative PCR (qPCR). With qPCR, minimal amounts of DNA can be amplified rapidly and detected easily. In contrast to classical PCR, qPCR can ascertain how much of a specific DNA sequence is present in a sample, rather than simply identify its presence or absence. Fuss et al. used this method to detect and quantify the abundance, identity, and disease transmission potential of intermediate host snails for intestinal schistosomiasis in Tanzania and detected infection in 35.4% of the snails [43].

#### 3.4.5. Molecular Barcoding

DNA barcoding is used for species identification. This is done by matching a standardized region of the genome—a barcode—from an unknown sample against all available barcodes in a reference library [48]. Amplification of different species-specific schistosome gene segments in snails, including the 18 S rDNA gene of *S.mansoni* and the DraI 121-bp repeat sequence of *S.haematobium*, has been tested successfully and applied in community studies on a large scale [39,49]. The mitochondrial gene COI has also been identified as a DNA barcode for cercarial detection. Levitz et al. [27] used the DNA barcoding approach to identify factors that determine distribution, abundance, and infection of *S.mansoni* with their genetic diversity.

#### 3.4.6. Loop-Mediated Amplification (LAMP)

LAMP is an isothermal nucleic acid amplification technique [50] that achieves high target specificity because of two sets of primers that span six distinct sequences of target DNA, i.e., LAMP experiments are designed with 4 to 6 primers that are specific to identified regions of target DNA or RNA sequences. A LAMP experiment results in rapid accumulation of double-stranded DNA and amplification byproducts that can be detected by a variety of methods. Abbasi et al. [38] developed two new LAMP assays to detect *S*. *mansoni* and *S*. *haematobium* DNA. Their results revealed that it was possible to detect infection from the first day after exposure to miracidia.

### 3.5. Theme 4: Environmental Diagnostics

Recently, eDNA has been used for schistosome species detection in African fresh waters [44]: Sengupta et al. [36], utilized the environmental DNA (eDNA) method that relies on screening water samples collected directly from the natural environment for schistosome surveillance. Alzaylaee et al. [37] designed a protocol where snails are collected and housed in experimental containers to allow them to shed cercariae, before eDNA in the water—from whole cercariae, cellular debris, or DNA chemically bound or in solution—is collected and its abundance measured using quantitative PCR. Alzaylaee et al. [44] developed a novel eDNA method that detects human Schistosoma in freshwater bodies. This they achieved by developing primers and probes for the mitochondrial 16 S rRNA region of human Schistosoma. This method was used in Tanzanian freshwater bodies to detect *S*. *mansoni* and *S*. *haematobium*.

## 4. Discussion

This review provides information on the challenges and opportunities presented by the current techniques for detecting schistosome infections in intermediate host snails. There is a paucity of literature on the comparison of each detection technique with regards to advantages and limitations, sensitivity, specificity, cost, and suitability of use in large surveys. The limited evidence is rather worrying considering the WHO plan to eliminate schistosomiasis as a public health problem, interrupt transmission in selected areas, and reduce the incidence of vector-borne diseases including schistosomiasis by at least 40% in 2025 [17,51].

The IHS of *S*. *haematobium*, *S*. *mansoni*, and *S*. *japonicum* are *Bulinus* spp., *Biomphalaria* spp. and *Oncomelania* spp., respectively. The presence of schistosomes in intermediate host snails indicates the infection prevalence in the environment. Hence, xenomonitoring is used to identify possible transmission sites, monitor administered interventions, and determine the efficacy of ongoing control programs. We adopted the WHO’s ASSURED criteria in the discussion section to identify the most appropriate diagnostics to use in detecting schistosome infections in IHS in a resource-constrained setting.

Findings from the review showed that the conventional diagnostics are the most used techniques for detecting schistosomes in intermediate host snails. According to WHO’s ASSURED criteria [21], the conventional techniques for identification of infected snails (shedding, crushing, or dissection of snails for sporocyst inspection) are simple, cheap, and relatively equipment-free. However, these methods are discounted on several fronts, mainly on account of their underestimation of the true prevalence of infection. Firstly, the methods are beset by low sensitivities, given that cercarial shedding can be highly focal and of low frequency, even in areas of substantial transmission [38]. Secondly, the method of cercariae shedding is inexpedient in detecting prepatent infection and delayed sporocyst development in cold seasons [40]. Conventional methods for cercarial detection cannot detect the parasite in dead infected snails or in the prepatent periods and are unable to speciate [40]. Thirdly, the labor-intensive nature of classic procedures, including the collection and maintenance of snails and the associated costs, are major hindrances in their application [52]. Even though the methods are generally affordable at initial stages of operation, they are time-intensive and high in operational costs at the end owing to the necessity for trained personnel, appropriate laboratory structures, and collection of snails in suspected areas as well as the maintenance and analysis of individual snails. Finally, cercarial identity from infected snails cannot be precisely detected morphologically to the species level. Without such precision, misleading epidemiological and transmission conclusions can be arrived at [53]. In concurrence with other study findings e.g., those of Abath et al. [52], Sengupta et al. [36], Weerakoon et al. [54], and El Einin et al. [41], our study found that the sensitivity of the classical approaches to cercariae detection is generally low because only a small percentage of infected snails are found even in areas with known high human infections.

Owing to the shortcomings of the conventional methods, suitable immunological methods of snail infection detection, such as the use of monoclonal antibodies (MAbs), have been presented as potentially more efficient detection tools in the field [55]. El Einin et al. [41] compared the superiority of immunodetection assay to PCR in detecting *S*. *mansoni* infection in *B*. *alexandrina*. Infections were detected in the second week and on the third day post-miracidial infection using immunodetection assay and PCR, respectively. A similar study was carried out by Hamburger et al. [56] comparing immunodetection assay to DNA probing technique for detecting infection considering the diagnostic qualities, technical aspects, cost of the test, and suitability of being applied in large-scale surveys. Immunodetection assay proved to be superior to the DNA probing technique. According to Einin et al. [41] the limitation of this technique is its non-ready availability for large-scale use because of the high cost of specific microsomal antigens used for antibody-capture. However, Hamburger et al. [56] shared a different opinion that the technique is cost-efficient.

Farghaly et al. [26] and Aboelhadid et al. [22] compared the conventional diagnostic, cercarial shedding by light exposure and snail crushing, to polymerase chain reaction (PCR) assay used in detecting infections in snails. Their results were consistent with that of Weerakoon et al. [54], which showed that PCR assay was more sensitive than the conventional methods because it could detect infections whereas the conventional methods could not. In addition, conventional methods are not suitable for large-scale surveys. Okeke et al. [42] employed PCR assay to detect schistosomes and nested PCR (nPCR) assay to characterize the schistosomes. Seventy-seven percent (77%) of the snails found were infected, with 9.67% of the snails positive for *S*. *mansoni* infection, the authors did not specify the species of the other infections. The results of Farghaly et al. [26], Aboelhadid et al. [22], and Okeke et al. [42] are consistent with those of Hamburger et al. [57] and Hanelt et al. [58], who used PCR and nPCR assays, respectively, to detect *S*. *mansoni* infections as early as one day post-exposure to a single miracidium. Hamburger et al. [57] concluded that PCR assay is suitable for detecting schistosome infections on a large scale.

Fuss et al. [43] used real-time PCR to determine the *S*. *mansoni* infection rate in Ijinga Island, Mwanza, north-western Tanzania to be 35.4%. However, they stated that the technique is expensive and is not suitable for field settings. Contrasting findings were reported by Kane et al. [59], who developed a new approach to detect and quantify schistosome DNA in freshwater snails using either fluorescent probes in real-time PCR or oligochromatographic dipstick assays targeting the ribosomal intergenic spacer and showed that this method is suitable for rich and constrained resource settings.

Further advances in the application of molecular based diagnostics in the detection of schistosomes in IHS have been achieved with LAMP assays. Abbasi et al. [38] developed two new LAMP assays that detected *S.mansoni* and *S.haematobium* infections from the first day after exposure to miracidia. They highlighted that LAMP assays as well as PCR assays are sensitive and specific techniques for detecting both prepatent and patent infection. However, PCR utilizes complex technology and needs to be performed by trained molecular biologists whereas the LAMP technique does not. In addition, the LAMP technique is suitable for field surveys. LAMP diagnostics are cheaper than PCR based diagnostics but more expensive than conventional diagnostics. Similar findings were obtained in the study by Hamburger et al. [60] that evaluated the suitability of the LAMP assay in detecting schistosome-infected field snails. Their results indicated that LAMP assays are suitable for detecting *S.haematobium* and *S.mansoni* in a low-technology parasitology laboratory. Local survey teams without experience in molecular biology acquired operational expertise for the first time in a few hours. Additionally, Qin et al. [61] evaluated the detection of *S.japonicum* infection in *Oncomelania hupensis* snails. LAMP was found to be superior to the conventional and PCR diagnostics. It is suitable for field conditions and requires less time. Infection rates of 0.4% and 6.5%, respectively, were obtained when the conventional and LAMP assays were used to detect infection. Finally, Gandasegui et al. [62] developed and evaluated the Biompha-LAMP assay to detect *S.mansoni* in experimentally infected snails as a diagnostic tool for field conditions. Results showed that the Biompha-LAMP assay is specific, sensitive, rapid, and potentially adaptable as a cost-effective method for screening of intermediate hosts infected with *S.mansoni* in both individual snails and pooled samples and is suitable for large-scale field surveys.

Evaluating the presence of cercariae directly from a water source is a useful way of identifying possible schistosome transmission sites. Theron et al. used a differential filtration technique to detect more than 80% of schistosome cercariae in turbid water. The apparatus used for this technique is light, easy to use even by unqualified personnel, and requires no power supply. Hence, it is suitable for field use [63]. Kloos et al., modified and tested the technique of differential filtration by Theron to detect *S.haematobium* cercariae in the laboratory and field in Upper Egypt. Most cercariae were recovered between 07:00–09:00 h [64]. Sengupta et al. [36] developed an environmental DNA (eDNA)-based tool to efficiently detect DNA traces of the parasite *Schistosoma mansoni* directly from freshwater samples and test its applicability at known transmission sites in Kenya. Comparison of schistosome detection by conventional diagnostics and eDNA showed 71% agreement between these methods. The eDNA method has higher sensitivity compared to the conventional diagnostic because it could detect schistosome presence at two sites where the conventional diagnostic failed. In addition, they argued that the overall cost-effectiveness of the eDNA method depends on the number of samples required. The cost of conventional diagnostics is similar to that of the eDNA method if few samples are required, but eDNA costs more when many samples are needed. However, the eDNA technique will need further refinement before it can be suitable for large-scale schistosomiasis surveillance and control programs. Alzaylaee et al., [37] reported on the xenomonitoring method that allows schistosome infections in host snail species to be determined from eDNA in water used to house those snails. This approach identified snails infected with *S.mansoni* and *S.haematobium*. They argued that this method produces more accurate and efficient results compared to the method used by Sengupta et al. [36] (direct eDNA from water) as extrinsic factors such as water flow and temperature could have an influence. It has the potential of overcoming potential taxonomic complications due to sympatric coexistence of human and non-human schistosome species with morphologically similar cercariae. Alzaylaee et al. [44] detailed an eDNA qPCR assay with high sensitivity and specificity that can detect schistosomes in freshwater intermediate host snails. The effectiveness of this novel technique was assessed by using it in eight freshwater sites in Tanzania. They detected schistosome presence where the conventional diagnostic approaches had failed. This is similar to the results of Sengupta et al. [36]. However, despite the accuracy of this technique in detecting infections, the sampling design, storage, and assay methods will need refinement for efficient results and affordable field utility.

This study was beset by some limitations. The review was limited to studies conducted in Africa and published in English. Hence, it is possible that some important articles might have been excluded. However, because Africa represents a large portion of resource-poor settings, the selected studies were deemed to be representative of the resource settings envisioned in this study. The study included only articles that focused on techniques used in detecting schistosome infection in human intermediate host snails. More articles could have been captured if language, country where study was conducted, and human intermediate host snails had not been determining factors.

## 5. Conclusions and Future Insights

This review found evidence that recent advances in technology have led to the discovery and innovation of novel techniques for cercariae detection in infected snails to understand schistosomiasis transmission dynamics. Most of these diagnostic methods hold promise for the improved management and control of the disease. The review discussed current cercariae detection methods and their appropriateness in resource-constrained settings. Many variables were found to influence the appropriateness of different detection techniques in different settings. Despite the high sensitivity and specificity recorded by the advanced immunological, nucleic-acid amplification, and eDNA diagnostics used in detecting schistosome infections in IHS, findings from the reviewed literature showed that conventional diagnostics are still the most used because they are affordable and easy to use. However, the insensitivity of the conventional diagnostics due to their inability to detect prepatent infections may, in combination with their labor-intensive nature, prevent the detection of many schistosomiasis infections in IHS in settings that have low schistosome prevalence. This poses a challenge to the goal of schistosomiasis control and elimination. On the other hand, although PCR-based diagnostic methods are highly sensitive and specific, they are not widely used in low-income countries due to the cost, highly technical requirements, and need for skilled personnel, making them unviable for routine application in field conditions. LAMP diagnostics, however, were found to be a highly field-suited tool and have been used in Africa. Still, the affordability of diagnostics that are identified as very sensitive, such as the PCR-based and LAMP methods, is a challenge for most resource-constrained settings. Hence, efforts should be made to design and develop new diagnostics that are high in specificity and sensitivity as well as affordable for use in such settings. In addition, to ensure that they are suitable for use in field laboratories and for large-scale surveys, these approaches should not be too technically demanding. This review found a need for more comprehensive and comparative evaluations of the diagnostic qualities and technicalities of all cercariae detection techniques as well as their feasibility for field and large-scale survey applications. Very few studies compared the various diagnostics in terms of their detection of schistosome infections. More importantly, we identified the need for cost-benefit analyses of each of these techniques.

## Figures and Tables

**Figure 1 ijerph-18-05403-f001:**
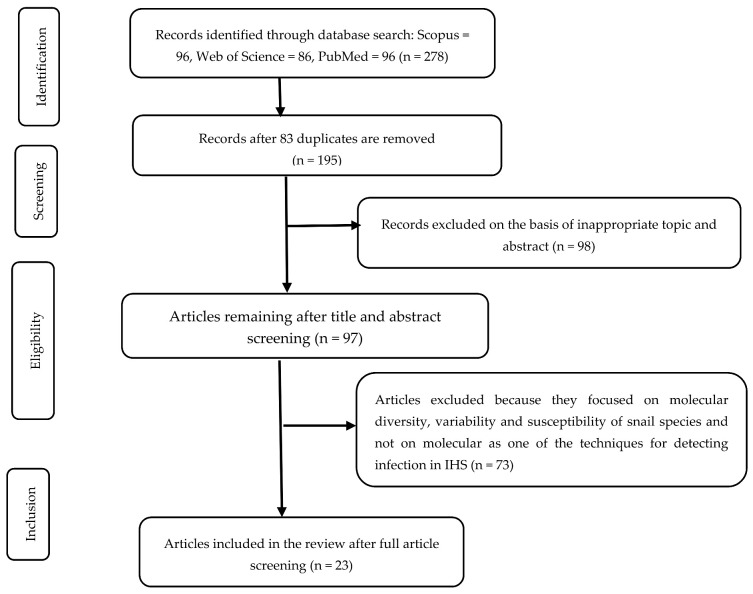
Preferred Reporting Items for Systematic Reviews and Meta-Analyses (PRISMA) flow diagram showing the process of article selection.

**Table 1 ijerph-18-05403-t001:** Summary of studies that were reviewed.

Author/Reference	Snail Species	Technique(s) and Theme Identified	Objective	Outcome	Advantages	Limitations	Country
Schols et al. [34]	*B. pfeifferi, B. globosus, B. africanus, L. natalensis*	Snail shedding, Rapid diagnostic multiplex PCR (RD-PCR): Infection RD-PCR and Schistosoma RD-PCR Conventional and nucleic-acid amplification diagnostics	To develop a two-step approach that detects trematode infections and discriminates *Schistosoma* spp.	High sensitivity and specificity of the multiplex RD-PCR approach. The assay is suitable for diagnostic screening of field-collected gastropods and species-level identification.	Results from the RD-PCR technique are more reliable than snail shedding because the technique is more sensitive and specific and can discriminate *Schistosoma* spp. The infection RD-PCR uses an internal control that excludes false-negative results to overcome the limitation of loop-mediated isothermal amplification (LAMP).	Not indicated.	South Africa and Zimbabwe
Bakuza et al. [24]	*Biomphalaria pfeifferi*	Cercarial shedding, snail dissection, and molecular approach (PCR and sequencing) Conventional and nucleic-acid amplification diagnostics	To detect the infection status and distribution of *Schistosoma mansoni* in snails using cercarial shedding and molecular approaches.	An infection prevalence of 12% was observed in snails based on cercarial shedding; 47% of the snails were PCR-positive. Sequence data were used for species identification.	PCR-based screening can be combined with sequencing to diagnose parasite infections and confirm the species of vectors and parasites interacting in a given region.	Cercarial shedding should not be relied on as it is less sensitive in detecting infection when compared to PCR.	Western Tanzania
Abu ElEinin et al. [41]	*Biomphalaria alexandrina*	Enzyme-linked immunosorbent assay (ELISA) and molecular detection of infected snails using nested PCR Immunological and nucleic-acid amplification diagnostics	To rapidly and accurately identify infected *B. alexandrina* snails.	The nested PCR assay provided higher efficiency for determination of infection prevalence in snails. It detects infection after 3 days post-infection whereas ELISA detects infection after the second week of miracidial infection.	They are time-efficient, and can be used for detecting infected snails on a large scale. High specificity for detection of prepatency in *B. alexandrina* snails infected with *S. mansoni*.	Although these methods are sensitive and specific, they are not easy to implement as they require trained personnel.	Egypt
Luka et al. [28]	*Bulinus globosus, Bulinus forskalli*, and *Lymnaeus natalensis*	Cercarial shedding Conventional diagnostics	To determine the distribution of snail intermediate hosts and infection rate in Borno State, Nigeria.	Three species of intermediate host snails were identified through shell morphology with an infection rate of 0.94%.	It is the commonly used method and is easy to use.	Cercarial shedding technique is affected by factors such as light and temperature. In addition, it is time- and energy-consuming as snails that did not shed cercariae on the day of exposure are re-exposed subsequently until the 7th day.	Nigeria
Abbasi et al. [38]	*Biomphalaria glabrata*	Loop-mediated isothermal amplification (LAMP) assays Nucleic-acid amplification diagnostics	To develop two LAMP assays for detecting *Schistosoma mansoni* and *S.haematobium* DNA in infected snails.	The developed assay is able to identify infections from the first day after exposure to miracidia.	Highly sensitive and specific in detecting schistosome infections. It is also suitable for large-scale monitoring of prepatent snail infection. Finally, it is not as expensive as PCR and can be used without having to train special cadres of molecular technologists.	The practical use of LAMP in field laboratories for diagnosis and monitoring schistosomiasis transmission requires further system development and validation. The LAMP assay is more expensive than the conventional methods and needs a molecular technologist to handle it.	Kenya
Aboelhadid et al. [22]	*Bulinus truncatus* and *Biomphalaria* *alexandrina* snails	Conventional techniques (cercarial shedding, snail crushing) and PCR. Conventional and nucleic-acid amplification diagnostics	To detect infections in *B.truncatus* and *B.alexandrina* snails and detect *S.mansoni* infection in *B.alexandrina* snails using PCR.	*B.alexandrina* snails shed *S. mansoni*, Pharyngeate longifurcate type I and type II cercariae while *B. truncatus* snails shed *S.haematobium* and Xiphidiocercaria cercariae. *B.alexandrina* snails, which shed *S.mansoni* cercariae in the laboratory, gave positive reaction in the samples. PCR detected latent infections in snails but the conventional techniques could not.	PCR possesses higher level of sensitivity in detecting infection than conventional techniques. The former has the capacity to detect prepatent infection while the latter does not. It is a potential tool that can be used in monitoring schistosome transmission on a large scale.	Results obtained when conventional techniques used in detecting schistosome infection are underestimated. Although PCR possesses higher sensitivity than conventional techniques, there are cases of inaccurate detection.	Egypt
Mereta et al. [29]	*B. pfeifferi, Biomphalaria sudanica, B. globosus, Bulinus forskalii*, and *Lymnaea natalensis*	Cercarial shedding Conventional diagnostics	To determine the distribution of IHS infection rates in the Omo-Gibe River Basin, southwest Ethiopia.	Eight cercariae types were identified morphologically from five freshwater snail intermediate host species with an infection rate of 3.6%, which is significantly lower than the 58% infection rate recorded in the study area.	Cercarial shedding and the use of morphological characteristics to identify snails and associated trematodes are cheaper than using molecular techniques.	Using cercarial shedding and morphological characteristics to identify snails and their associated trematodes is not as efficient and accurate as using molecular techniques.	Ethiopia
Sengupta et al. [36]	*Biomphalaria glabrata*, *Biomphalaria pfeifferi*	Cercarial shedding and environmental DNA (eDNA) Conventional and nucleic-acid amplification diagnostics	To develop an environmental DNA (eDNA) based method for schistosome detection in aquatic environments.	The suitability of this method was tested at known transmission sites in Kenya. Comparison of schistosome detection by conventional snail surveys (snail collection and cercariae shedding) with eDNA (water samples) showed 71% agreement between the methods. However, eDNA method detected schistosome presence at two additional sites where snail shedding failed, demonstrating a higher sensitivity of eDNA sampling.	eDNA has a higher sensitivity than conventional methods for detecting infection. In addition, the cost of eDNA is comparable to that of cercariae shedding if few samples are required. Furthermore, it detects schistosome larval stages infection directly from freshwater hence, eliminating the need for extensive snail surveys and it is good for low intensity infection areas.	eDNA is more expensive than cercariae shedding if many samples are required. In addition, it is not suitable for large-scale schistosomiasis surveillance and control programs as its validity under different field and habitat conditions is yet to be accessed. The eDNA filter capture method needs to be fine-tuned as there are difficulties in filtering turbid water.	Kenya
Alzaylaee et al. [37]	*Bulinus globosus* and *Biomphalaria pfeifferi*	Cercarial shedding, environmental DNA (eDNA) and quantitative PCR (qPCR). Conventional, eDNA and nuclei-acid amplification diagnostics.	To report a xenomonitoring method that allows schistosome infections of host snail species to be determined from eDNA in water used to house those snails.	eDNA accurately diagnosed the presence of *S. mansoni*-infected snails in all tests, and *S. haematobium*-infected snails in 92% of tests. In addition, the abundance of *Schistosoma* eDNA in experiments was directly dependent on the number and biomass of infected snails.	The eDNA-based xenomonitoring method overcame the challenge of blocked filters when filtering turbid waters, which could lead to false negatives by allowing sediments to settle before filtering.	The effectiveness of the eDNA- based xenomonitoring method for schistosome hosts other than *Bulinus globosus* and *Biomphalaria pfeifferi* is unknown. The eDNA-based xenomonitoring method involves a lot of processing and is time-consuming.	Tanzania
Levitz et al. [27]	*Biomphalaria stanleyi, Biomphalaria sudanica* and *B. pfeifferi*	Cercarial shedding and DNA barcoding Conventional and nucleic-acid amplification diagnostics	To identify the factors that determine distribution, abundance, and infection of *Biomphalaria* spp.	Snail infection was found to be positively correlated with electrical conductivity while snail abundance was correlated with low pH. DNA barcoding revealed a complex heterogeneous landscape of cercariae.	Not indicated	Not indicated	Uganda
Saad et al. [33]	*B. alexandrina, B. truncatus, Lymnaea natalensis*, *Melanoides tuberculata*	Snail crushing and multiplex PCR Conventional and Nuclei-acid amplification diagnostics.	To evaluate and identify the different trematode infections in four snail species.	The multiplex PCR based on ITS-1 region was used to identify *B. alexandrina**, B. truncatus, Lymnaea natalensis*, and *Melanoides tuberculata*.	Molecular based methods have higher levels of sensitivity and specificity for detecting infection in digenean trematodes than the conventional techniques. They are useful for large-scale screening of snail populations.	The snail crushing method is not capable of detecting prepatent infections.	Egypt
Opisa et al. [32]	*Biomphalaria sudanica, Biomphalaria pfeifferi*, and *Bulinus globosus*	Cercarial shedding Conventional diagnostics	To determine the presence of intermediate host snails and to ascertain whether active transmission was occurring within the area.	High abundance of *Biomphalaria* and *Bulinus* spp. shedding cercariae confirmed the existence of the risk of schistosomiasis transmission within the area.	Easy to use and more cost-effective compared to molecular methods.	Parasite prevalence could be severely underestimated when cercarial shedding technique is used, since it cannot detect prepatent infections. Schistosome species specific to humans can only be identified and qualified using molecular methods.	Western Kenya
Fuss et al. [43]	*Biomphalaria* species	Real-time polymerase chain reaction Nucleic-acid amplification diagnostics	To determine the abundance, identity, and disease transmission potential of intermediate host snails for intestinal schistosomiasis.	The *Biomphalaria* snail species found in areas with human water contacts were infected with Schistosoma. Hence, there is a high risk for schistosomiasis transmission at most water contact points around Ijinga Island.	More sensitive in detecting infections than cercarial shedding.	The technique is expensive and cannot be used directly in field settings for routine assessment of Schistosoma infection in snails. Further testing is required to rule out cross-reactions with other trematodes. Furthermore, distinctions cannot be made between pre-patent and patent infections.	Western Tanzania
Farghaly et al. [26]	*Biomphalaria alexandrina*	Conventional methods (cercarial shedding and snail crushing) and PCR Conventional and nucleic-acid amplification diagnostics	To evaluate a PCR assay used in detecting *S. mansoni* infection in *B. alexandrina* and to compare the PCR method of infection detection to the conventional detection methods.	PCR is superior to the conventional methods and can detect positive cases that were negative when examined by shedding or crushing methods.	PCR had an average 100% sensitivity and specificity. PCR technique could detect prepatent infection.	The cost of reagents needed for PCR is relatively high when compared to the conventional methods. PCR methods requires skilled and trained technicians. The crushing and shedding methods had sensitivity of 46.4% and 23.8%, respectively, compared to the PCR method.	Egypt
Akinwale et al. [39]	*Bulinus* species	PCR amplification of the Dra1 repeat Nucleic-acid amplification diagnostics	To identify *Bulinus* snails involved in the transmission of *S. haematobium* in Nigeria.	Out of 138 *Bulinus globosus and Bulinus truncatus* snails collected from 28 study sites, 41 (29.7%) were positive for schistosome infection.	Sensitive in detecting if a snail is infected or not.	It cannot identify the parasite species the snail is infected with. It could result in partial, ambiguous profiles, where it was unclear as to whether the snails were only lightly infected or if there had been miracidial penetration of the snail surface but no actual development of the parasite.	South-west Nigeria
Okeke et al. [42]	*Biomphalaria* spp	Nested polymerase chain reaction (nPCR) methods Nucleic-acid amplification diagnostics	To detect schistosome infections and identify the species of *Biomphalaria* spp. found in the geographical area.	High prevalence of schistosome infection with 164 snails found infected out of 212 snails that were screened. Larger number of schistosome-infected snails were detected using PCR compared to research conducted earlier in the same study area that used the traditional method.	Higher sensitivity compared to the crushing method and conventional PCR.	The nPCR technique is more expensive compared to the conventional method. The nPCR technique is specific to *S. mansoni*.	Nigeria
Amarir et al. [40]	*Bulinus truncatus*	Molecular approach: DraI PCR and Sh110/Sm-Sl PCR Nucleic-acid amplification diagnostics	To estimate the snail infection rates at the last historic transmission sites of schistosomiasis, known to be free from new infection among humans since 2004.	All snails from the five historic endemic provinces were negative for *S.haematobium* infection, supporting the absence of haematobium focus.	Is sensitive in detecting snails infected with *S. haematobium* from very early prepatency. PCR is practical for large-scale monitoring of *S. haematobium* transmission in affected communities.	The use of two PCRs may take more time and products. A drawback of using Sh110 SmSl PCR alone or DraI PCR alone in a co-endemic area is that they would not be able to detect mixed populations of *S. haematobium* and *S. bovis.* DraI PCR and Sh110 SmSl assays are very expensive.	Morocco
Alzaylaee et al. [44]	*Biomphalaria* spp., *Bulinus* spp. and *Oncomelania* spp.	qPCR-based environmental DNA (eDNA) assay Nucleic-acid amplification diagnostics	To report a diagnostic tool that can identify DNA from three human-infecting Schistosoma species within water samples.	The eDNA assay can detect schistosomes in freshwater bodies.	The eDNA avoids the requirements to locate, identify, and individually test the infectious status of host snails. A more sensitive and specific technique compared to the conventional technique.	There are shortcomings in the sampling design, storage, and assay methods. The utility of eDNA assays is dependent on the nature of the schistosome life stages present in the environment. The eDNA assay is not reliable in detecting infections with low eDNA concentrations. The success of this technique is dependent on large snail abundance and high infection prevalence.	Tanzania
Ejehu et al. [25]	*Lymnaea*, *Bulinus* and *Pila*.	Cercarial shedding Conventional diagnostics	To identify the intermediate host snails responsible for schistosomiasis transmission.	More snails were recorded in the rainy season than dry season. Irrespective of the dredging activities in the lake, the snail host species was present, indicating the risk of urinary schistosomiasis infection in the community, although none of the snails found shed cercariae.	The cercarial shedding technique is easy to implement and more cost-effective.	The cercarial shedding technique is unable to detect prepatent infections, which can last for several weeks with only a proportion of snails reaching the stage of cercarial shedding.	Nigeria
Senghor et al. [35]	*B. senegalensis* and *B. umbilicatus*	Cercarial shedding Conventional diagnostics	To determine and ascertain the role of identified freshwater snail species in the transmission of *S. haematobium* and study the impact of drought on the snails.	The infection rate was 0% in 2012 and 0.12% in 2013, respectively, for *B.senegalenis*, and 13.79% in 2012 and 4.98% in 2013, respectively, for *B. umbilicatus*. Snails between 7–9.9 mm were the sizes most commonly collected in July for both snail species. Adult *B. umbilicatus* and *B. senegalensis* can withstand 7 to 8 months of drought.	Not indicated	Not indicated	Senegal
Moser et al. [31]	*Bulinus truncatus*, *Bulinus forskalii*, and *Biomphalaria pfeifferi*	Cercarial shedding Conventional diagnostics	To determine the spatio-temporal distribution of schistosomiasis intermediate host snails and to assess *Schistosoma* infection in snails.	413 *Bulinus truncatus*, 369 *Bulinus forskalii*, and 108 *Biomphalaria pfeifferi* snails were collected and subjected to cercarial shedding. In the dry season, 1 *B. truncatus* out of 119 (0.84%) snails and 1 *B. pfeifferi* out of 108 (0.93%) snails collected shed *Schistosoma* spp. cercariae. The snails collected after the rainy season did not shed *Schistosoma* spp. cercariae. The abundance of *B. truncatus* and *B. forskalii* showed an inverse U-shape relationship with the square term of conductivity. *B. pfeifferi* showed a negative, linear association with pH in the dry seasons.	Not indicated	Not indicated	N’Djamena, Chad
Mohammed et al. [30]	*B. pfeifferi*, *B. truncatus, B. forskalli, Cleopatra bulimoides, M. tuberculata, Physa acuta*, and *Lymnea natalensis*	Cercarial shedding Conventional diagnostics	To explore the coexistence of freshwater snail species in relation to environmental factors.	10,493 snails from seven species were collected. 48.6% were *B. pfeifferi* species. 14.1% of the snails found shed some type of cercariae. *B. truncatus* species was heavily infected with 46.2% prevalence. Double infections were recorded in two *B. truncatus* snails and one *Cleopatra bulimoides* snail. Snail densities were lower during the summer months compared to the winter months. *M. tuberculata* snails were not affected by seasonal changes.	Not indicated	Not indicated	East Nile locality, Khartoum state, Sudan
Allan et al. [23]	*Bulinus globosus* and *B. nasutus*	Cercarial shedding and PCR using DraI repeat Conventional and nucleic-acid amplification diagnostics	To investigate the seasonal variation and relationships between snail abundance, *S. haematobium* patency rates, prepatent exposure and force of transmission in the field.	Within field-collected snails, a large difference was noted in the prevalence of prepatent *S. haematobium* infections (detected by PCR) compared to patent infections. The numbers of snails with prepatent infections were approximately ten-fold higher.	PCR approach can detect prepatent infections whereas cercarial shedding cannot.	Although PCR detects prepatent infections, these infections often fail to develop to patency. This may be due to the incompatibility of host and parasite combinations, the longevity of the snails, or a combination of these effects.	Tanzania

## Data Availability

Not applicable.

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
