# Peer review of "Challenges and Opportunities Presented by Current Techniques for Detecting Schistosome Infections in Intermediate Host Snails: A Scoping Review"

_ijerph, 2021, doi:10.3390/ijerph18105403_

Round 1

Reviewer 1 Report

The review written by Nwoko et al., is interesting and it was carried out following the PRISMA guidelines.

However, there are few points that require attention:

  1. Under Theme 3 (line 260) is better to classify this category as “Nucleic acid amplification methods”, since in general terms, the PCR-based (classical PCR, nested PCR multiplex PCR and real-time quantitative PCR) are considered different from the LAMP.
  2. The phrase in lines 294-296 is confusing; you should clarify that 77% were positive for Schistosoma, but only 9.67% were positive for S. mansoni. Using the term “infected” is misleading…infected with what?
  3. The references should be carefully checked; some of them do not include the manuscript tile, other lack page numbers (e.g. #15 and #16) and some are just cited wrong (#4 and #17 cite the WHO as Organization, W.H.) (this is not a comprehensive list, so please check them all).
  4. Link to references #50 and #38 are duplicated (lines 337 and 342, respectively).
  5. Although I understand that figures improve the visual aspect of a manuscript, Figures 2 and 3 in this case are not really necessary; Figure 2’s information is replicated in the text and Figure 3 only shows the general scheme of a PCR. If you want to keep one, at least you should modify Figure 3 to depict how all the techniques mentioned work.
  6. The limitation described for LAMP on Table 1 for the study by Abbasi (ref #38) probably was true when the paper was published in 2010, but I believe that it is no longer true…costs have dropped dramatically.

Reviewer 2 Report

All comments are in the attached document.

Reviewer 3 Report

In the present study, the authors performed a systematic review of the literature to assess the advantages and limitations of different diagnostic techniques for detecting schistosomes infections in snails. Literatures from Scopus, Web of Science, and PubMed databases from 2008 to 2020 were searched using combinations of predefined search terms with  Boolean operators. The review is well designed and the used citations covered the study scope. Before accepting for publication, minor revision and modification of the article in light of the appended comments need to be addressed.

  1. Pls, check the journal guidelines for the limits of keywords number
  2. please change all parasites' names “plasmodium, toxoplasma…….” into italic through the text.
  3. L79-99, please move to the study limitation
  4. Pls summarize your important findings that you mentioned in the conclusion in a figure
  5. Pls, add a section in the review entitled “New insights into the future of detecting schistosome infections in intermediate host snails” and clarify in it your insights to overcome the limitations of different diagnostic techniques for detecting schistosomes infections in snails depending on the review findings.
